# Dietary Folic Acid Supplementation Attenuates Maternal High-Fat Diet-Induced Fetal Intrauterine Growth Retarded via Ameliorating Placental Inflammation and Oxidative Stress in Rats

**DOI:** 10.3390/nu15143263

**Published:** 2023-07-24

**Authors:** Huaqi Zhang, Xinyu Zhang, Yutong Wang, Xuenuo Zhao, Li Zhang, Jing Li, Yabin Zhang, Peng Wang, Hui Liang

**Affiliations:** Department of Nutrition and Food Hygiene, School of Public Health, Qingdao University, 308 Ningxia Road, Qingdao 266071, China; huaqi_erin@163.com (H.Z.); zzzzzxy1022@126.com (X.Z.); wyt687199@126.com (Y.W.); 19916466685@163.com (X.Z.); zhangli200513@163.com (L.Z.); lijing_95123@163.com (J.L.); zhangyabin77@163.com (Y.Z.); wpeng@qdu.edu.cn (P.W.)

**Keywords:** fetal intrauterine growth restriction, placenta, folic acid, inflammatory responses, oxidative stress, placental dysfunction

## Abstract

The placenta is particularly susceptible to inflammation and oxidative stress, leading to placental vascular dysfunction and placental insufficiency, which is associated with fetal intrauterine growth restriction (IUGR). It is unknown whether folic acid (FA) supplementation can alleviate high-fat diet-induced IUGR in rats by improving placental function. In this study, pregnant rats were randomized into one of four diet-based groups: (1) control diet (CON), (2) control diet supplemented with FA, (3) high-fat diet (HFD), and (4) high-fat diet supplemented with FA (HFD + FA). Dams were sacrificed at gestation day 18.5 (GD18.5). The results indicated that dietary FA supplementation normalized a maternal HFD-induced decrease in fetal weight. The decrease in placental efficiency, labyrinth zone (LZ) area, blood sinusoid area, vascular density, and the levels of angiogenesis factors induced by a maternal HFD were alleviated by the addition of FA, suggesting that FA supplementation can alleviate placental vascular dysplasia. Furthermore, FA supplementation increased the protein expressions of SIRT1, inhibited NF-κB transcriptional activation, attenuated the levels of NF-κB/downstream pro-inflammatory cytokines, induced Nrf2 activation, and increased downstream target protein expression. In conclusion, we found that dietary FA supplementation during pregnancy could improve maternal HFD-induced IUGR by alleviating placental inflammation and oxidative stress, which may be associated with the regulation of SIRT1 and its mediated NF-κB and Nrf2 signaling pathways.

## 1. Introduction

Intrauterine growth restriction (IUGR) affects up to 8% of pregnancies worldwide, is a main hazard factor for perinatal mortality and morbidity, and may contribute to metabolic disorders in adulthood [1,2]. Studies have shown that poor food choices among mothers can elevate the risk of IUGR [3,4]. In addition, the adverse impacts of a maternal high-fat diet (HFD) on the intrauterine environment and fetal growth have been demonstrated in numerous animal studies [5,6]. Placenta is an active participant in pregnancy and plays a central role in determining fetal growth [5]. Placental insufficiency is considered to be an important predictor of IUGR and mediates, in part, the impacts of maternal HFD on negative fetal outcomes [7]. The evidence suggests that maternal HFD-induced lipotoxicity leads to placental inflammation and oxidative stress [8,9], which may impair placental development and vascular function, and ultimately affect fetal growth and development [10].

Folic acid (FA), a water-soluble B-complex vitamin, is an indispensable nutrient for fetal growth and development [11,12,13,14]. FA supplementation during pregnancy is recommended in most countries to reduce the incidence of fetal neural tube defects (NTDs) [15]. Recently, studies have shown that FA consumption during pregnancy results in placental metabolic and angiogenesis gene expression changes [16]. A retrospective population-based cohort study showed that FA antagonist exposure was associated with adverse pregnancy outcomes mediated by the placenta [17]. Animal experiments showed that FA may prevent lipopolysaccharide-induced preterm delivery, fetal death, and IUGR [18]. Nevertheless, it remains obscure whether FA can alleviate maternal HFD-induced IUGR by modulating placental function.

FA has a critical role in energy metabolism and mitochondrial function through the regulation of sirtuin-1 (SIRT1) expression in tissues and cells [19,20]. SIRT1, a member of the nicotinamide adenine dinucleotide (NAD)-dependent protein deacetylase family, has been shown to be expressed in placental tissues [21]. Accumulating evidence suggests that SIRT1 regulates the inflammatory cytokines related to placental endothelial cell homeostasis via inhibiting NF-κB transcriptional activity [22,23]. On the other hand, SIRT1 is thought to increase resistance to oxidative damage by activating the downstream target Nrf2 and moving it to the nucleus, where it initiates antioxidative gene expression [24]. Furthermore, Nrf2/HO-1 activated in human placental trophoblasts cells reduces the production of the anti-angiogenic factor soluble fms-like tyrosine kinase-1 (sFlt-1) and improves endothelial dysfunction [25]. However, whether FA can against maternal HFD-induced placental inflammation and oxidative stress and alleviate IUGR by upregulating SIRT1, thus inhibiting NF-κB and activating the Nrf2 pathway, has not been reported.

In this work, female Sprague Dawley (SD) rats were fed an HFD during gestation and FA supplementation 2 weeks before mating and throughout gestation to observe the impacts of maternal dietary FA supplementation on IUGR and to explore its possible mechanisms associated with the placenta.

## 2. Materials and Methods

### 2.1. Animals and Experimental Design

SD rats were purchased from Vital River (Beijing, China) at 8 weeks of age and weighed 250 ± 10 g. Animals were maintained under a specific pathogen-free (SPF) environment. After 1 week of adaptive feeding, female rats were randomly assigned to receive either a normal FA diet (2 mg FA/kg) or an FA-supplemented diet (5 mg FA/kg). After 2 weeks, female rats in both groups mated with male rats in a ratio of 3:1. Gestation day0.5 (GD0.5) refers to the day when a sperm-positive vaginal smear was found. On GD0.5, the pregnant rats in the two FA diet groups were allocated to one of two dietary feeding groups, respectively. In short, four diets were designed to study the effects of two different levels of FA (2 and 5 mg FA/kg) during pregnancy, both in the presence and absence of HFD. The four experimental groups were: control diet (CON, *n* = 8), control diet supplemented with FA (FA, *n* = 8), high-fat diet (HFD, *n* = 8), and high-fat diet supplemented with FA (HFD + FA, *n* = 8). Animal groupings and dietary composition are depicted in Figure 1A. Details on dietary composition are provided in Appendix A. Throughout the study, water and food were freely available to all animals. Body weights were measured weekly.

Gestation rats were weighed, fasted for 6 h, and anesthetized on GD18.5. Maternal blood, placentas, and fetuses were collected. The litter was considered as the unit for the statistical comparison of fetal and placental parameters among the four groups. For fetal weight, placental weight, fetal crown-rump length, and placental diameter, the means were calculated per litter and then averaged per group. In the subsequent detection of placental-related indicators, placental samples were randomly selected from different litters. The whole study design is portrayed in Figure 1B.

All animal experiments were approved by the Animal Care and Use Committee of the medical college of Qingdao University and strictly complied with the institution’s guidelines on laboratory animals (approval number: QDU-AEC-2023346).

### 2.2. Histological Analysis of Placenta

Placental tissue was fixed in 4% paraformaldehyde and dehydrated through a gradient of ethanol and embedded. The tissues were sectioned (5 μm) on a tissue microtome. Subsequently, they were stained with hematoxylin and eosin (HE). Scans were carried out using a 3DHISTECH Panoramic SCAN slide scanner, and the images were analyzed using Panorama Viewer software (version 1.0.7). Placental composition and the area of blood sinusoids were assessed using ImageJ (version 1.48).

### 2.3. Maternal Serum Biochemical Assays

The levels of triglyceride (TG), total cholesterol (TC), high-density lipoprotein (HDL)-cholesterol, and low-density lipoprotein (LDL)-cholesterol were measured using an automated biochemical analyzer.

Serum insulin was measured using an enzyme-linked immunosorbent assay (ELISA) kit (Mercodia AB, Uppsala, Sweden). The Homeostasis Model Assessment of Insulin Resistance (HOMA-IR) index is computed as (fasting plasma insulin [mIU/L] × fasting plasma glucose [mmol/L])/22.5.

### 2.4. Placental Biochemical Assays

The supernatant obtained by homogenizing placental tissue and centrifuging at 2500 rpm for 15 min was used to measure the concentration of placenta growth factor (PLGF), sFlt-1, vascular endothelial growth factor A (VEGF-A), vascular endothelial growth factor receptor 2 (VEGFR2), vascular cell adhesion molecule 1 (VCAM-1), interleukin-1β (IL-1β), interleukin 6 (IL-6), tumor necrosis factor-α (TNF-α), Cyclooxygenase-2 (COX-2), malondialdehyde (MDA), glutathione peroxidase (GSH-Px), superoxide dismutase (SOD), and catalase (CAT) using ELISA kits (Jiancheng Technology, Nanjing, China).

### 2.5. Western Blotting Analysis

Western blotting followed our previous procedures [26,27,28]. For nuclear extraction, placental whole lysates were suspended in a hypotonic buffer and stored on ice for 15 min. The suspension was then detergent-mixed and centrifuged for 30 s at 14,000× *g*. The concentration of protein was determined using BCA kits (P0012S, Beyotime, Shanghai, China). For immunoblots, the proteins were separated by 10% SDS and transferred to a PVDF membrane (Millipore, Billerica, MA, USA). Membranes were blocked with 10% skimmed milk in Tris-buffered saline/Tween (TBST) and incubated with primary antibodies for 12 h. The expression of the following proteins in the placental homogenates was determined by the use of antibodies: SIRT1, NF-κB p65, Nrf2, p-IκB, IκB, NQO1, and HO-1 (Cell Signaling Technology, Danvers, MA, USA). Lamin B was used as the loading control for the nuclear protein. β-actin was used as the loading control for the total protein. Subsequently, the membranes were washed with TBST for 10 min each, followed by incubation with the secondary antibody (Cell Signaling Technology, Danvers, MA, USA) for 2 h.

### 2.6. Immunohistochemistry

Placenta sections were dewaxed and rehydrated through an ethanol gradient elution, followed by overnight incubation with anti-CD31 (BD Biosciences Pharmingen, San Diego, CA, USA). After washing in TBS, tissues were incubated with the corresponding secondary antibody for 2 h at RT, followed by a reaction with diaminobenzidine (DAB) and counterstained with hematoxylin. A digital tissue biopsy scanner was used to acquire images of the sections. ImageJ (version 1.48) was used to quantify the CD31 integrated optical density (IOD).

### 2.7. Statistical Analysis

Statistical analysis was conducted with SPSS 22.0 (SPSS, Chicago, IL, USA) and GraphPad Prism 8.0 (GraphPad, San Diego, CA, USA). All data were presented as the mean ± SD. The differences between the groups were analyzed using a one-way analysis of variance (ANOVA) supplemented by a posthoc test for the least significant difference (LSD) to determine the difference between the two groups. *p* < 0.05 was regarded as statistically significant.

## 3. Results

### 3.1. Maternal Body Weight, Liver Weight, and Adipose Tissue Weight at GD18.5

The rate of weight gain of the pregnant rats was remarkably higher in the HFD group than that in the CON and FA groups during gestation, whereas the gestational body weight gain rate in the HFD + FA group was significantly diminished when compared to the HFD group (*p* < 0.05; Figure 2B). Figure 2C–F showed that the liver weight, liver index, intraperitoneal fat weight, and intraperitoneal fat index significantly increased in the HFD group when compared to the CON and FA groups; the HFD + FA group had markedly lower liver weight, liver index, intraperitoneal fat weight, and intraperitoneal fat index when compared to the HFD group (*p* < 0.05).

### 3.2. Maternal Serum Biochemical Indexes at GD18.5

As Figure 3A shows, the serum level of TG was significantly higher in the HFD group than those in the CON group (*p* < 0.05). In comparison to the HFD group, the HFD + FA group had a markedly lowered serum level of TG (*p* < 0.05). The maternal serum levels of TC, HDL-C, LDL-C, glucose, and insulin showed no difference among the four groups (*p* > 0.05, Figure 3B–F). The HOMA-IR in the HFD group was significantly higher than the CON and FA groups (*p* < 0.05, Figure 3G).

### 3.3. Maternal FA Supplementation Ameliorates Fetal and Placental Development at GD18.5

At GD18.5, there were no differences between the four groups in litter size, fetal crown-rump length, placental weight, or placental diameter (*p* > 0.05; Figure 4A,C–E). Notably, the HFD group had a lower fetal weight and placental efficiency than the CON and FA groups (*p* < 0.05). Moreover, fetal weight and placenta efficiency were more elevated in HFD + FA than in the HFD group (*p* < 0.05; Figure 4B,F). The placental TG and TC levels were significantly higher in the HFD group than in the CON and FA groups (*p* < 0.05). FA supplementation significantly reduced the TG and TC levels when compared to the HFD group (*p* < 0.05; Figure 4G,H). The histomorphological analysis of the placentas showed that the HFD group had a reduced placental area and labyrinth zone (LZ) area when compared to the CON and FA group, but the LZ area was elevated in HFD + FA when compared to the HFD group (*p* < 0.05; Figure 4J,K).

### 3.4. Maternal FA Supplementation Ameliorates Placental Angiogenesis at GD18.5

When compared to the CON and FA groups, the blood sinusoidal area and CD31 integrated optical density (IOD) were significantly reduced in the HFD group (*p* < 0.05). The blood sinusoidal area and the IOD of CD31 were significantly elevated in the HFD + FA group when compared to the HFD group (*p* < 0.05; Figure 5A–D).

Placental levels of PLGF, VEGF-A, and VEGFR2 were significantly elevated in the HFD group when compared to the CON and FA groups, while the levels of sFlt-1, sFlt-1/PLGF, and VCAM-1 were remarkably elevated (*p* < 0.05; Figure 6A–F). Placental levels of PLGF, VEGF-A, and VEGFR2 were noticeably elevated in the HFD + FA group than those in the HFD group, while the levels of sFlt-1, sFlt-1/PLGF, and VCAM-1 were markedly reduced (*p* < 0.05).

### 3.5. Maternal FA Supplementation Ameliorates Placental Inflammation and Oxidative Stress

When compared to the CON and FA groups, the IL-1β, IL-6, TNF-α, and COX-2 levels in the placentas of the HFD group were significantly elevated (*p* < 0.05), the (above four) levels in the HFD + FA group were remarkably diminished in the HFD group (*p* < 0.05; Figure 7A–D). The placental concentration of MDA increased significantly, and SOD, CAT, and GSH-Px activity was significantly lower in the HFD group than in the CON and FA groups (*p* < 0.05). When compared to the HFD group, the MDA concentration had markedly decreased, and the activities of SOD, CAT, and GSH-Px had significantly increased in the HFD + FA group (*p* < 0.05; Figure 7E–H).

### 3.6. Effects of Maternal FA Supplementation on SIRT1-Mediated Inflammatory and Oxidative Stress Signaling Pathway

The expression level of SIRT1 in the HFD group was lower than in the CON and FA groups. The expression level of SIRT1 was noticeably higher in the HFD + FA group than in the HFD group (*p* < 0.05; Figure 8A). Moreover, the levels of Nucl-p65/Cyto-p65 and p-IkBa/IkBa were higher in the HFD group than in the CON and FA groups, while they were significantly lower in the HFD + FA group than in the HFD group (*p* < 0.05; Figure 8B). Similarly, the expression of Nucl-Nrf2/Cyto-Nrf2, NQO1, and HO-1 was lower in the HFD group when compared to the CON and FA groups, whereas they were noticeably lower in the HFD + FA group than in the HFD group (*p* < 0.05; Figure 8C).

## 4. Discussion

Few studies have demonstrated the protective effect of FA on IUGR by regulating placental function. For the first time, our study found that maternal dietary FA supplementation ameliorated HFD-induced placental dysfunction by alleviating inflammation and oxidative stress, thereby improving IUGR.

IUGR is defined as the failure of the fetus to achieve genetic growth capacity [29]. HFD is a common type of poor food choice among reproductive-age women. The adverse effects of an HFD before and during pregnancy on fetal growth and development have been well documented [9,30]. Several animal studies of pre-pregnancy maternal HFD have shown that maternal obesity or metabolic disorders adversely affect fetal outcomes, including IUGR [9,30,31,32,33]. However, in real life, a proportion of women begin to change their dietary structure after pregnancy, with the aim of providing more nutrition for their fetuses [34]. Increasing studies have found that HFD exposure during pregnancy alone can also cause lower fetal body weight [35,36]. In this study, we intervened with an HFD during pregnancy in rats and found their fetal weights at GD18.5 were significantly lower when compared to fetuses from the CON-fed dams, indicating the occurrence of IUGR.

Moderate supplementation using specific nutrients during pregnancy may decrease the risk of IUGR [37]. A prospective study conducted on more than 6000 pregnant women demonstrated that periconceptional FA supplementation (0.4–0.5 mg/d) leads to higher placenta and birth weight [38]. Similarly, a cohort study from China indicated that a maternal FA supplementation of 400 µg per day reduced the risk of small for gestational age (SGA) [39]. These findings suggest that appropriate FA supplementation during the gestation period may promote fetal growth. In animal studies, orally administered FA (3 and 15 mg/kg) has been shown to prevent lipopolysaccharide-induced preterm delivery and IUGR [18], but it has not been reported whether FA can improve IUGR induced by maternal HFD during pregnancy. Dietary doses of 2 mg/kg and 5 mg/kg were commonly applied in rodent FA supplementation studies because these concentrations correspond to the recommended FA intake level for humans and the tolerable upper FA intake level for humans, respectively, which could help with better extrapolating our research results to humans [40,41]. In this study, SD rats were fed an HFD while 5 mg/kg of FA was added to the diet, and we found that FA supplementation significantly normalized the HFD-induced fetal weight loss in rats. This is the first evidence that FA supplementation alleviates IUGR caused by maternal HFD.

The growth of the fetus is reliant on the placenta; therefore, placental dysfunction is one of the major reasons for IUGR [22,42]. The ratio of fetal/placental weight indicates placental development, and the function of nutrient transportation is commonly used to evaluate the efficiency of the placenta [43]. Our study found that dietary FA supplementation improved the HFD-induced decline in placental efficiency, although no remarkable differences in the weight of the placentas were detected between the four groups. Similar results can be seen in other studies, where the placenta morphology may have been altered, although the weight did not change [44]. The placenta is composed of three zones: the decidual zone (DZ), junctional zone (JZ), and labyrinth zone (LZ) [45]. The LZ is the closest layer to the fetus, consisting of the maternal sinusoids and trophoblastic and fetal capillaries, which grow rapidly during the last week of pregnancy to support fetal growth [46]. When compared to the other regions of the placenta, the LZ is the layer where oxygen and nutrients are exchanged between the mother and fetus, which is more susceptible to injury due to high blood flow and active cell proliferation [47]. In this study, the morphology of the placentas showed that the placental area and LZ area from the HFD dams had been markedly reduced. In pregnancies supplemented with FA, the increase in placental area occurred mainly in the LZ, which is explains well the associated increase in fetal growth. Maternal HFD consumption causes the proportion of the blood sinus area in the LZ to be significantly reduced; this, conversely, can lead to impaired nutrient transport in the placenta. However, dietary FA supplementation can restore the loss of blood sinusoid area caused by a maternal HFD. Consequently, improvements in placental morphology and blood sinusoidal area may explain the improvement in fetal weight in the HFD + FA group.

The development of placental blood vessels is an important factor in determining placental function [48]. CD31 is a sensitive marker of angiogenesis. The immune-histochemical analysis of placental CD31 staining showed that maternal HFD severely reduced the IOD of CD31 in the LZ. This is supported by a recent study on lower angiogenesis in the placenta of HFD-fed dams, which was associated with IUGR [33]. In our study, maternal FA supplementation significantly increased placental angiogenesis. Furthermore, placental angiogenic factors, including pro-angiogenic factors and anti-angiogenic factors, participate in the complex process of placenta angiogenesis and are essential for the successful establishment of early angiogenesis in the placenta [49]. VEGF-A binds to its receptor VEGFR2, which is expressed in endothelial cells to induce angiogenesis. PlGF, which belongs to the angiogenic VEGF family, acts by enhancing the action of VEGF-A. sFlt-1, as a decoy receptor, binds to VEGF-A and PlGF, causing the reduced bioavailability of each cell to its target cells [50]. An imbalance between pro-angiogenic factors and anti-angiogenic factors (i.e., an increased sFlt-1/PlGF ratio) leads to a net anti-angiogenic condition that promotes the occurrence of IUGR [49]. Our results show that the pro-angiogenic factors in the placenta of HFD rats are downregulated, while the anti-angiogenic factors are upregulated, and maternal FA supplementation can reverse the imbalance between them. The results are in accordance with previous studies, indicating that the improvement in placental efficiency may be due to better angiogenesis [51]. Consequently, it can be believed that maternal FA supplementation improves IUGR by increasing placental angiogenesis.

Sirtuins represent a NAD-dependent deacetylases family that has recently been shown to be regulated by FA [19,20]. SIRT1 plays a significant regulatory role in DNA damage repair, anti-apoptosis, anti-aging, and energy metabolism, and it is worth mentioning that it exerts anti-inflammatory and antioxidant effects in multiple tissues by inhibiting inflammatory responses and inducing antioxidant defense pathways [52]. The placenta of pregnant women on HFDs shows lipid accumulation, leading to placental inflammation and oxidative stress, which is recognized as the most vital event in placental functional injury, such as vascular dysfunction [8]. In this study, increased TG and TC levels in the HFD group placentas suggested that maternal HFD-induced placental lipid accumulation and maternal FA supplementation may improve HFD-induced placental lipotoxicity. SIRT1 has been shown to be highly expressed in trophoblasts. SIRT1-null mice had smaller placentas and abnormalities in both the LZ and JZ [53]. Another study showed that maternal overnutrition reduces SIRT1 expression in placental tissue [54]. Thus, it can be speculated that SIRT1 may be a key player in the protection of FA against placental dysfunction. In our study, a maternal HFD decreased placental SIRT1 protein expression, while FA supplementation upregulated SIRT1 protein expression. Besides, SIRT1 activation has been shown to downregulate NF-κB p65 transcriptional activity and positively modulate Nrf2 antioxidant signaling [55], suggesting that the FA-upregulation of SIRT1 in the placenta may inhibit NF-κB p65 and activate the Nrf2 pathways against inflammatory and oxidative stress responses.

Inflammation and oxidative stress cause placenta dysfunction, which prevents the effective transfer of nutrients to the fetus [56]. Studies have shown that HFD-induced placental inflammation causes an imbalance in placental angiogenesis [48]. The transcription factor NF-κB p65, a main mediator of inflammation, is regulated by SIRT1. Under unstimulated conditions, NF-κB p65 usually combines with the inhibitor of kappa B alpha (IκBα) and is maintained in the cytoplasm. IκBα phosphorylation leads to the nuclear translocation of NF-κB p65, which then stimulates pro-inflammatory cytokines synthesis [57]. In this study, we found that FA activated placental SIRT1 expression, thereby inhibiting IκBα phosphorylation and NF-κB p65 nuclear translocation, ultimately downregulating pro-inflammatory cytokines, including TNF-α, IL-6, IL-1β, and COX2. In fact, a growing number of studies indicate that FA has anti-inflammatory effects [58]. In accordance with our findings, another study indicated that FA against LPS-induced IUGR by inhibiting COX-2 expression in the placenta [18]. Moreover, placenta oxidative stress induces vasoconstriction and pathologic vascular remodeling, which are related to the pathophysiology of IUGR [59]. Placenta is vulnerable to oxidative stress because of the high metabolic activity of placental cells. Placental oxidative stress is a prominent feature of pregnancy disorders, including IUGR [60]. The increased concentration of lipid peroxidation products (MDA) in the HFD group indicates that the placenta was damaged by oxidative damage caused by the maternal HFD. SIRT1 protects cells against oxidative stress damage via its downstream regulator, Nrf2. Nrf2 is activated and transferred to the nucleus, where it instills antioxidant genes and antioxidant enzyme transcription, leading to the attenuation of oxidative damage [61]. Our results demonstrated that FA leads to increased SIRT1 expression and ultimately promoted Nrf2 nuclear translocation and upregulated the expression of HO-1 and NQO1 in the placenta. Some studies have illustrated the potential effect of Nrf2/HO-1 activation in reducing sFlt-1 production and improving endothelial dysfunction [25,62]. In our study, the improvement in placental sFlt-1 levels in the HFD + FA group may be associated with the activation of Nrf2. Moreover, FA supplementation also increased the activities of the Nrf2-regulated antioxidant enzymes SOD, CAT, and GSH-Px, further corroborating the mitigation of placental oxidative impairment. These findings jointly suggest that FA supplementation attenuated the maternal HFD-induced placental inflammatory response and oxidative damage via inhibiting the NF-κB inflammatory path and promoting the Nrf2 antioxidant pathway via upregulating SIRT1 and, to some extent, ameliorates endothelial dysfunction and increases placental angiogenesis.

This study has a few limitations. First, the assessment at a single time point (GD18.5 in late pregnancy) cannot reflect the impact of FA supplementation on the dynamic changes in the placenta during pregnancy. Second, it has been suggested that maternal FA supplementation has sex-specific effects on the offspring [16,63,64], but we did not differentiate between the sex of the fetuses and placentas in this study. Third, considering the differences in species between humans and rats, the results obtained in this study cannot be directly extrapolated to the population. This study is the first to examine the effect of maternal FA supplementation on HFD-induced IUGR. Based on this study, more in-depth research can be conducted in the future. It can longitudinally examine the effects of FA on HFD-induced placental dysfunction by setting more time points during gestation and further linking the sex of the fetus. In addition, we can attempt to obtain more evidence from population studies on FA supplementation during pregnancy being able to improve IUGR.

## 5. Conclusions

This study showed that FA supplementation ameliorated HFD-induced placental dysfunction by regulating the SIRT1-mediated pathways, including the Nrf2 antioxidant pathway, and inhibiting the NF-κB inflammatory response, thereby improving IUGR. Maternal FA supplementation may serve as a new prophylactic strategy to prevent IUGR in humans.

## Figures and Tables

**Figure 1 nutrients-15-03263-f001:**
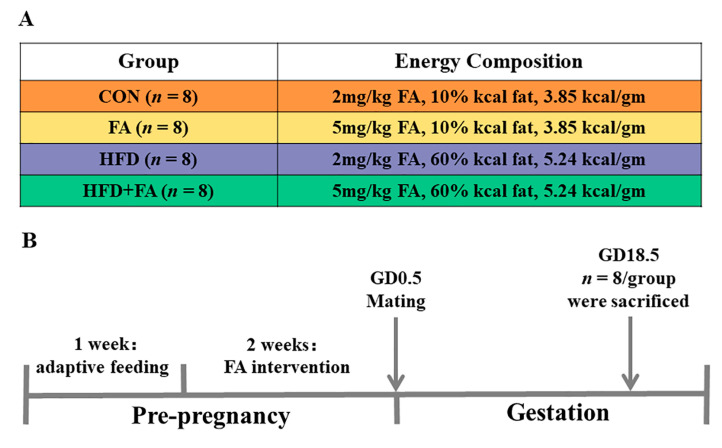
Design of the study. (**A**) Grouping of the study. (**B**) Flow chart of the study.

**Figure 2 nutrients-15-03263-f002:**
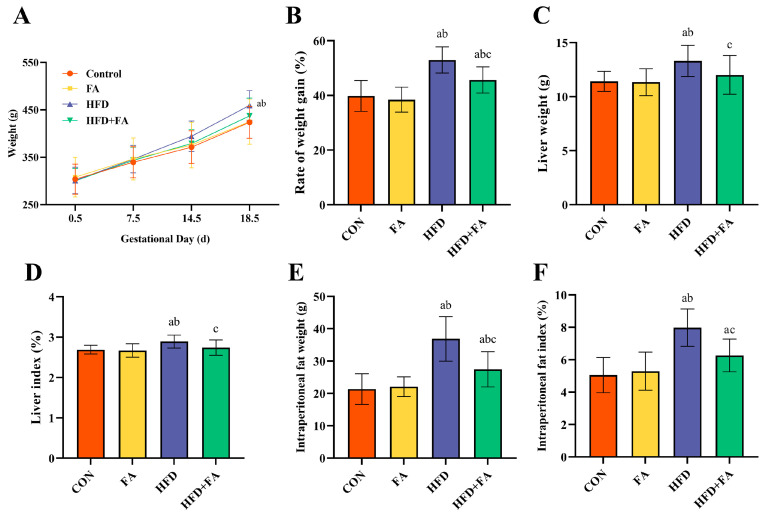
Maternal body weight, liver weight, and adipose tissue weight. (**A**) Maternal body weight. (**B**) Rate of body weight gain. (**C**) Liver weight. (**D**) Liver index. (**E**) Intraperitoneal fat weight. (**F**) Intraperitoneal fat index. Data are represented as means ± SD (*n* = 8). A significant difference (*p* < 0.05) is denoted by different letters: a vs. the CON group; b vs. the FA group; c vs. the HFD group.

**Figure 3 nutrients-15-03263-f003:**
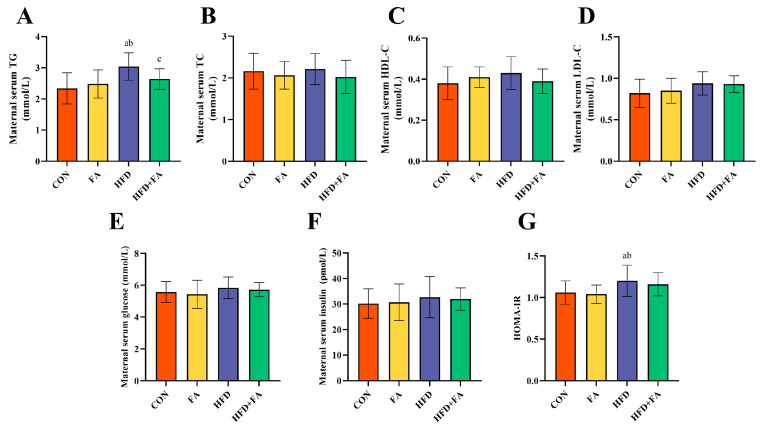
Maternal serum biochemical index. (**A**) Serum TG; (**B**) Serum TC; (**C**) Serum HDL-C; (**D**) Serum LDL-C; (**E**) Serum glucose; (**F**) Serum insulin; (**G**) HOMA-IR; data are represented as means ± SD (*n* = 8). A significant difference (*p* < 0.05) is denoted by different letters: a vs. the CON group; b vs. the FA group; c vs. the HFD group.

**Figure 4 nutrients-15-03263-f004:**
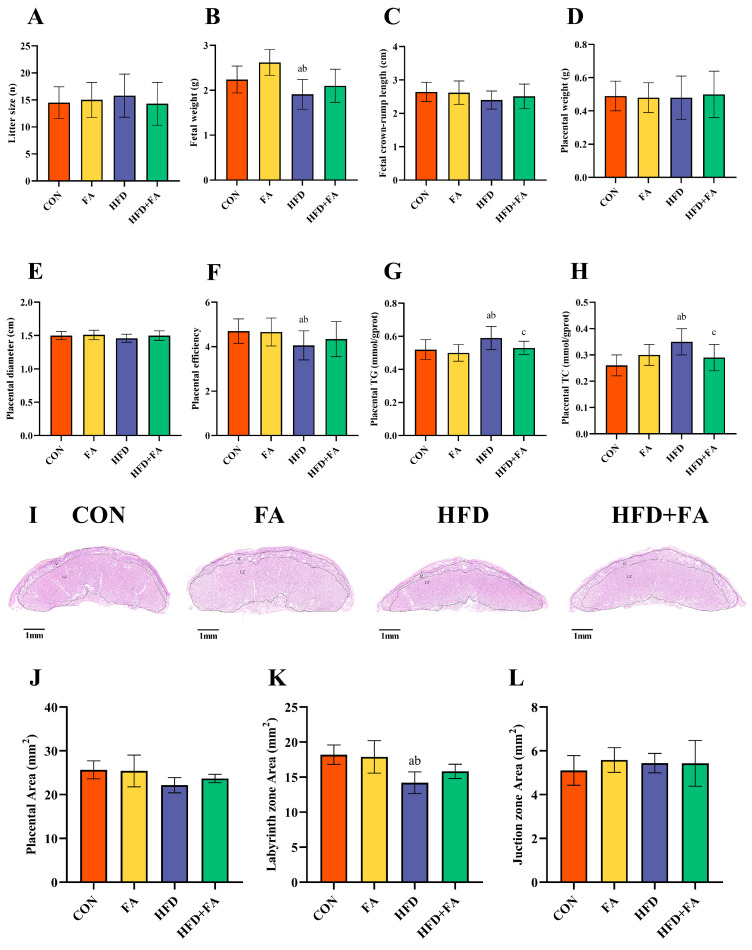
Fetal and placental development. (**A**) Litter size. (**B**) Fetal weight. (**C**) Fetal crown-rump length. (**D**) Placental weight. (**E**) Placental diameter. (**F**) Placental efficiency (fetal/placental weight ratio). (**G**) Placental TG. (**H**) Placental TC. (**I**) The morphology of the placentas; scale bar 1 mm; LZ, labyrinth zone; JZ, junction zone. (**J**) Placenta area. (**K**) LZ area. (**L**) JZ area. Data are represented as means ± SD (*n* = 8). A significant difference (*p* < 0.05) is denoted by different letters: a vs. the CON group; b vs. the FA group; c vs. the HFD group.

**Figure 5 nutrients-15-03263-f005:**
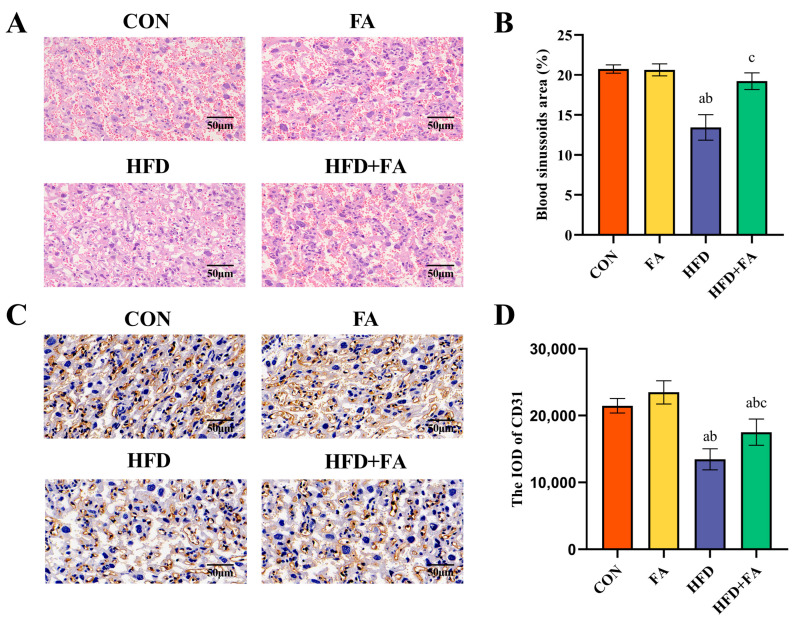
Morphological observation of the placenta. (**A**) Placental cross-sections stained with H&E. (**B**) Blood sinusoids area percentage. (**C**) Immuno-histochemical analysis for CD31 staining (brown). (**D**) The IOD of CD31 in LZ. Data are represented as means ± SD (*n* = 8). A significant difference (*p* < 0.05) is denoted by different letters: a vs. the CON group; b vs. the FA group; c vs. the HFD group.

**Figure 6 nutrients-15-03263-f006:**
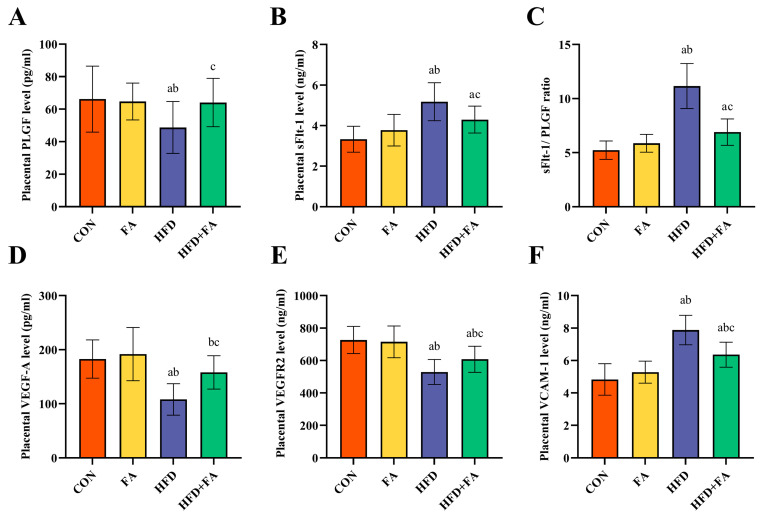
Functional parameters of the placenta. (**A**) Placental PLGF. (**B**) Placental sFlt-1. (**C**) Placental sFlt-1/PLGF ratio. (**D**) Placental VEGF-A. (**E**) Placental VEGFR2. (**F**) Placental VCAM-1. Data are represented as means ± SD (*n* = 8). A significant difference (*p* < 0.05) is denoted by different letters: a vs. the CON group; b vs. the FA group; c vs. the HFD group.

**Figure 7 nutrients-15-03263-f007:**
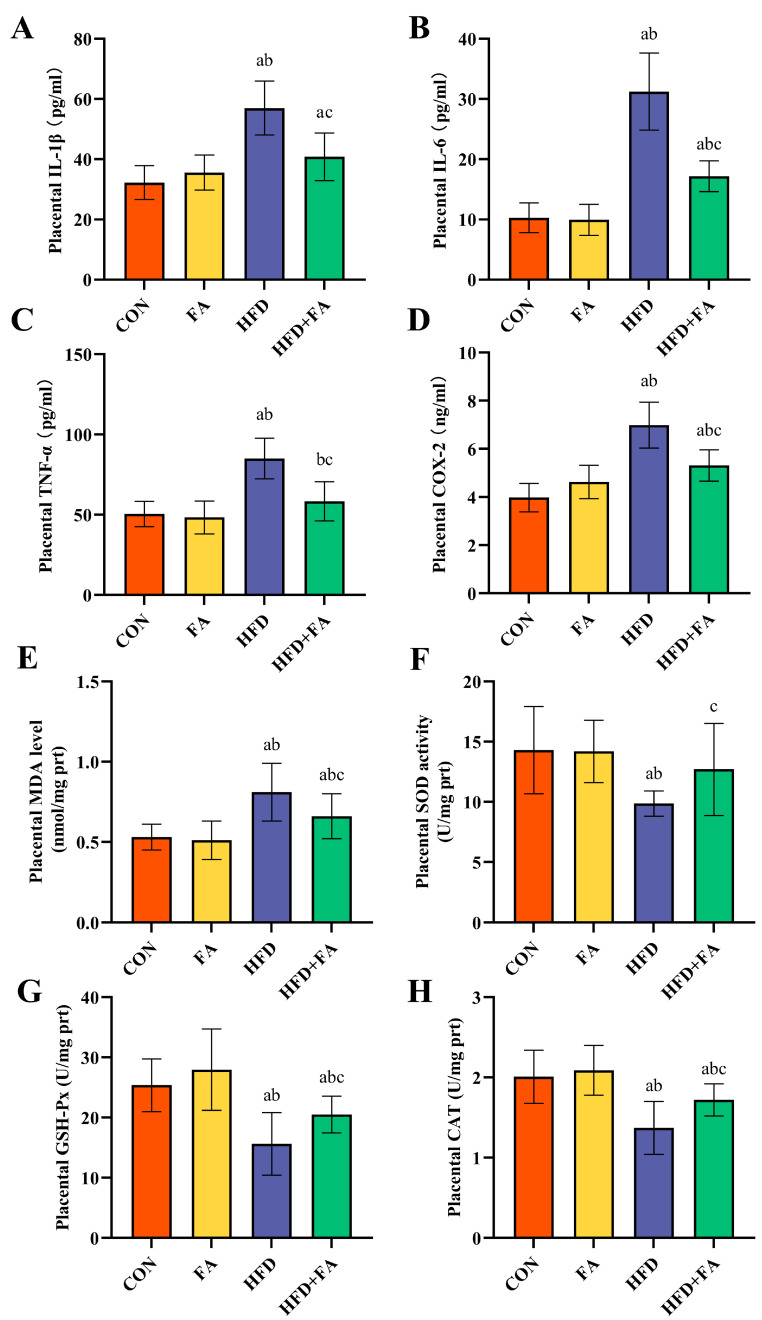
Inflammatory cytokine and oxidative stress parameter levels of the placentas. (**A**) Placental IL-1β. (**B**) Placental IL-6. (**C**) Placental TNF-α. (**D**) Placental COX-2. (**E**) Placental MDA. (**F**) Placental SOD. (**G**) Placental GSH-Px. (**H**) Placental CAT. Data are represented as means ± SD (*n* = 8). A significant difference (*p* < 0.05) is denoted by different letters: a vs. the CON group; b vs. the FA group; c vs. the HFD group.

**Figure 8 nutrients-15-03263-f008:**
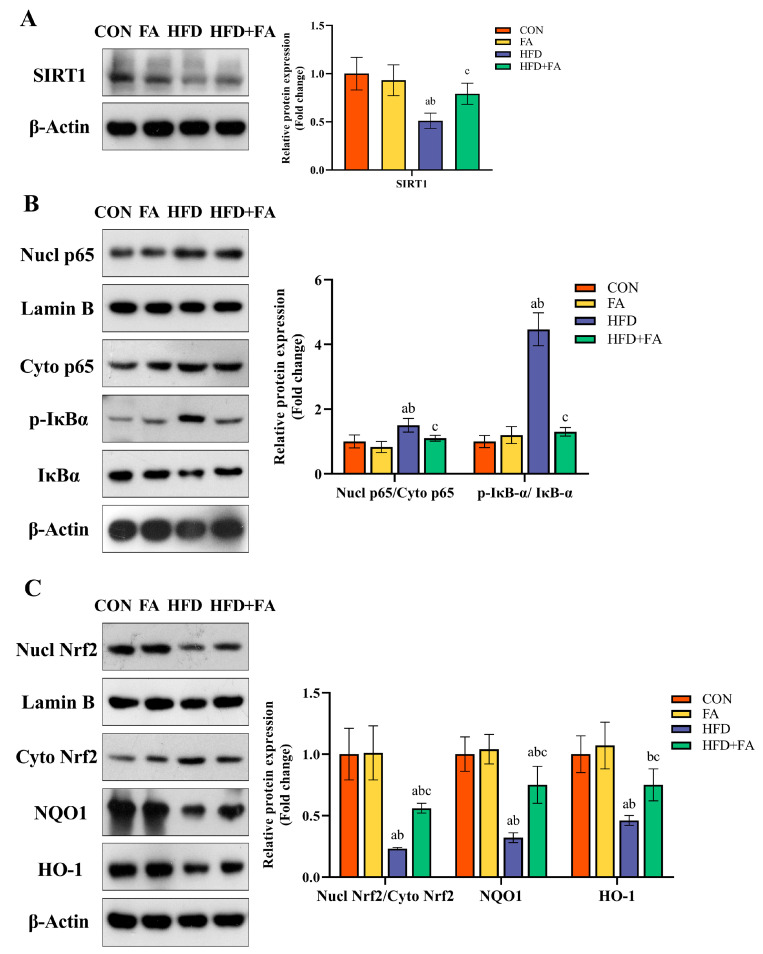
SIRT1/NF-κB and SIRT1/Nrf2 signaling pathway in the placenta. (**A**) The protein expression of SIRT1. (**B**) The protein expression of the SIRT1/NF-κB signaling pathway. (**C**) The protein expression of the SIRT1/ Nrf2 signaling pathway. Data are represented as means ± SD (*n* = 3). A significant difference (*p* < 0.05) is denoted by different letters: a vs. the CON group; b vs. the FA group; c vs. the HFD group.

## Data Availability

The datasets used and/or analyzed during the current study are available from the corresponding author upon reasonable request.

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
