# Peer review of "Dietary Folic Acid Supplementation Attenuates Maternal High-Fat Diet-Induced Fetal Intrauterine Growth Retarded via Ameliorating Placental Inflammation and Oxidative Stress in Rats"

_nutrients, 2023, doi:10.3390/nu15143263_

Round 1

Reviewer 1 Report

In this report, Zhang and co-workers explored the beneficial effects of dietary folic acid supplementation on maternal HFD induced IUGR through anti-inflammation and anti-oxidative stress effects in the placenta. The results are clear, however, a number of questions should be addressed.

1. Placental angiogenesis develops during different phases of gestational period, only one time point (GD18.5) was analyzed. It will be more convincing to test placental angiogenesis at different time points.

2. In the OGTT test, the dams were fasted for 12h, this seems to constitute a very long fasting period for a pregnant rodent, which could affect fetal growth and development. Also, long time fasting and OGTT challenge will stress the dams, which may affect the final results. Please justify.

3. How many placentas were collected per litter and how many were analyzed?

4. The author should determine the placental and fetal sexes. Placentas from which gender were used in this study? Sex-specific effects were observed in placentas in studies of folic acid supplementation (PMID 35268026; 34010503; 36515682).

5. Overnight fasting before sacrifice is relative long for pregnant rats, and fasting affects oxidative stress markers. It makes it hard to interpret the result of oxidative stress signaling pathway at GD18.5.

6. The HFD used most often in rodents frequently reduce fetal growth, while fetal overgrowth is typical feature found in obese human pregnancies. The authors should discuss this discrepancy.  

Author Response

请参阅附件。

Reviewer 2 Report

It is noteworthy that the study design considered delivering the high-fat diet once pregnant and not previously, given that there is evidence that the time of exposure to high-fat diet influences the results. It is recommended that they incorporate some justification for the decision of this type of experimental design.

Author Response

请参阅附件。

Round 2

Reviewer 1 Report

The authors responsed porperly. No more questions.